# Effects of Two Short-Term Aerobic Exercises on Cognitive Function in Healthy Older Adults during COVID-19 Confinement in Japan: A Pilot Randomized Controlled Trial

**DOI:** 10.3390/ijerph19106202

**Published:** 2022-05-19

**Authors:** Atsuko Miyazaki, Takashi Okuyama, Hayato Mori, Kazuhisa Sato, Keigo Kumamoto, Atsushi Hiyama

**Affiliations:** 1Information Somatics Laboratory, Research Center for Advanced Science and Technology, The University of Tokyo, Tokyo 153-8904, Japan; gtokuyama2000@yahoo.co.jp (T.O.); hiyama@star.rcast.u-tokyo.ac.jp (A.H.); 2Computational Engineering Applications Unit, Head Office for Information Systems and Cybersecurity, RIKEN, Saitama 351-0198, Japan; 3Department of Physical Therapy, Faculty of Health Sciences, School of Medicine, Kobe University, Kobe 650-0017, Japan; 4Super Reha, LLC., Tokyo 198-0074, Japan; morihaya49@gmail.com; 5Care 21 Co., Ltd., Osaka 530-0003, Japan; kazuhisa09@gmail.com; 6General Education Center, Nagano University of Health and Medicine, Nagano 381-2227, Japan; kumamoto.keigo@shitoku.ac.jp; 7Center for the Promotion of Social Data Science Education and Research, Hitotsubashi University, Tokyo 186-8601, Japan

**Keywords:** aerobic exercise, cognitive function, COVID-19, dancing, Nordic walking, older adults, physical function, body composition

## Abstract

Aerobic exercise improves executive function—which tends to decline with age—and dual-task training with aerobic exercise improves the global cognitive function. However, home-based older adults could not follow these programs due to social isolation during the coronavirus disease 2019 pandemic. Therefore, we conducted a single-blind randomized controlled trial with 88 healthy older adults without dementia or sarcopenia who were randomly assigned into the Nordic walking (aerobic exercise), dance (dual-task training with aerobic exercise), or control group. The participants in both exercise intervention groups trained for 30 min, three times per week, for 4 weeks. All groups consumed amino acid-containing foods three times per week. We found that both exercise intervention groups showed improvements in executive function, while the dance group showed additional improvement in global cognitive function. The dance group showed a higher maximum gait speed, greater improvement in imitation ability, and improved executive function and cognitive function than the Nordic walking group. The intervention programs did not significantly affect the muscle mass or muscle output than the control group; however, both programs improved the participant neurological functions such as the heel lift, with dance training being the most effective intervention. In conclusion, dance training effectively improves cognitive function.

## 1. Introduction

The proportion of the world’s population aged ≥ 60 years is increasing swiftly and is predicted to increase from 12% to 22% between 2015 and 2050, with the number already being >30% in Japan [1]. As life expectancy increases, the risk of age-related conditions such as chronic diseases may also increase [2]. Even with normal aging, changes occur in the neural basis of cognition, particularly affecting the prefrontal cortex, a region primarily involved in executive functions and attention, memory, and work [3,4]. This effect is greater in individuals with mild cognitive impairment (MCI) [5]. The prognosis of MCI is associated with a higher risk of developing Alzheimer’s disease (AD) [6] and other forms of dementia [7]. In Japan, dementia has been the most common reason for requiring care since 2016 [8]. Therefore, maintaining and improving the cognitive function is important in preventing dementia and the need for further care [9,10] and is important for successful aging [2]. Physical exercise is an effective way to improve cognitive function; it increases the functional capacity, cardiorespiratory function, muscle mass, and modulates neurotrophin activity in the brain [11]. For example, exercise increases the production of brain-derived neurotrophic factors, resulting in the generation of new neurons and the improvement in connectivity between the existing neurons [12,13]. Exercise-induced neuroplasticity also improved the brain volume, memory, and executive function in randomized controlled trials (RCTs) conducted among healthy older participants at risk of AD [14,15,16,17,18]. Additionally, aerobic exercise is strongly recommended (grade B evidence), especially in the Japanese medical guidelines, for preventing dementia [19]. It can also facilitate the reversal of hippocampal volume loss [20] and improve the executive function in older adults [21], increase the neural activity in the lateral frontal and parietal regions [22], and increase the prefrontal and temporal gray matter volume [22,23]. Furthermore, dual-task training with aerobic exercise improves the executive function and general cognitive function in older adults with and without cognitive impairment [24,25]. In a recent study, the authors observed improvements in attention and executive function and overall cognitive function as a result of physical exercise. Interestingly, they found that the relative telomere length increased in these participants compared to the control group after 6 months [26]. Thus, it is suggested that an exercise program that can efficiently improve age-related executive function and overall cognitive function may be a useful strategy for slowing down aging in general.

However, the semi-state-of-emergency measures enacted against the coronavirus disease 2019 (COVID-19) by the Japanese government prevents the cognitive and physical health of older adults; this is a consequence of the stringent distancing, quarantine, and isolation rules, which mainly affect older adults who have a high mortality rate due to the virus where age-related comorbidities can further increase the risk of death [27]. A previous review highlighted the importance of prescribing home-based exercise as a form of physical activity to older adults during COVID-19 [28]. Without effective strategies to maintain physical activity, there may be a detrimental impact on the public health infrastructure. Despite the urgent need for such strategies, the optimal program for improving cognitive function through physical exercise remains to be established. However, we can develop a program to prevent secondary problems consequent to the stringent measures against the COVID-19 pandemic. By selecting and delivering a self-directed home physical exercise program that is effective at improving cognitive function in healthy older individuals, we can develop a program to reduce the secondary problems associated with the measures imposed due to the COVID-19 pandemic.

Nordic walking (NW) is a form of aerobic exercise that engages the whole body through a physical activity program for older adults [29]. It is a recommended physical activity for this population because it does not aggravate shoulder and arm pain, or leg disabilities [30]. Therefore, based on previous studies, NW can improve executive function and gait in older adults, rendering it an at home physical-activity program feasible for healthy older adults during the COVID-19 pandemic.

Dance is also a form of aerobic exercise with physical benefits such as improved gait and balance [31], proposed for frail and sick older adults [32,33,34]. A recent integrative review reported functional and structural improvements in neuroplasticity in healthy older adults after dance training [35]. Dance is a dual-task activity that combines the cognitive, motor, and affective tasks through multiple components such as spatial awareness, movement coordination, balance, endurance, and interaction [36]. However, most studies using dance programs have been conducted outside Japan and vary from national and contemporary dances to ballroom and line dances.

Choreography refers to the sequence of steps and movements integrated to a dance. Learning choreography requires a memory mechanism that is gradually incorporated such as when learning movements using DVD images of the dancers [37,38,39,40]. This process directly involves the observation of the behavior of exemplary dancers and the imitation of their movements [41,42]. Although many previous studies have reported broad cognitive improvements [43], there are no specific reports on the learning effects of dance imitation. We believe that the imitation ability deserves attention in older adults because it is reportedly related to MCI and dementia [44] and can influence the ability to learn new motor programs in the future.

However, exercise intervention in older adults should be accompanied by nutritional intervention. In young adults, the maintenance of muscle mass involves a balance between muscle protein synthesis and degradation, whereas in older adults, muscle protein synthesis in response to dietary protein intake is impaired [45]. Therefore, muscle wasting may occur in older adults in response to exercise loads of 5.2 metabolic equivalents for NW and moderate- or higher-loads for dance, depending on the program. Thus, muscle protein degradation exceeds the synthesis, leading to decreased muscle mass [46]. Furthermore, the COVID-19 pandemic has restricted older adults’ access to food, leading to muscle mass loss caused by a potentially poor-diet quality [47].

Therefore, we conducted a three-group RCT using two exercise programs: outdoor NW, which can reliably provide aerobic exercise of a certain intensity (5.2 METs), and an indoor dance program, a dual-task exercise involving aerobic exercise. We evaluated the effect of physical activity on the cognitive and physical function in older adults during the COVID-19 pandemic.

## 2. Materials and Methods

### 2.1. Trial Design and Setting

This RCT was registered at the University Hospital Medical Information Network Clinical Trial Registry (UMIN000038740). Before enrollment, written informed consent for participation was obtained from each participant, and all study procedures adhered to the tenets of the Declaration of Helsinki. The study protocol was approved by the Ethics Committee of RIKEN in Saitama (approval number: Wako3 2019-28(3)). The CONSORT checklist for this study was included as a Appendix A.

We conducted a single-blind RCT with three parallel groups: a NW training group (walking group), an original dance program training group (dance group), and a protein-only group (control group). All groups received protein supplementation, but the control group received no exercise intervention. Randomization was performed using dynamic allocation according to the dates when the participants provided their informed consent. Research personnel who assessed the cognitive and physical functions were blinded to the group assignments. The program recruiters were blinded to the background or psychological profiles of the participants, and they did not have specific psychological aims. The Consolidated Standards of Reporting Trials statement [48] was used as the framework. The trial design is presented in Figure 1.

### 2.2. Participants

Due to the COVID-19 pandemic, the participants were recruited via the online bulletin boards of the Tokyo and Kanagawa Prefectures, from 10 October 2020 to 10 November 2020. Native-Japanese speakers with no history of dementia or diseases known to affect the central nervous system, heart, or respiratory system were included in the study if they were healthy community-dwelling adults aged ≥ 60 years without COVID-19 during the study. The exclusion criteria were vision or hearing impairments, regardless of corrective lenses or hearing aids. No previous experience with NW or dancing was necessary. In total, 90 participants were registered and provided informed consent. The Mini-Mental State Examination (MMSE-J) [49] was used to assess the degree of cognitive functioning; participants with an MMSE-J score < 24 points were excluded. A physical therapist screened all of the participants’ lower extremity joints for deformities, pain complaints, and range of motion to determine whether continued exercise was possible. Two patients were excluded because they did not adhere to the study schedule (Table 1).

### 2.3. Sample Size

There was no pre-determined sample size for this pilot study as no previous studies have conducted an intervention test for the cognitive and physical function levels with three groups. Furthermore, given the COVID-19 pandemic, this study enrolled individuals who wanted to participate.

### 2.4. Overview of the Interventions

All interventions were conducted at the participants’ homes. Each program was performed 3 days per week for 4 weeks, according to the participant’s schedule. The participants spread out their training days throughout the week. The participants were required to record notes on their training and protein intake on scheduled days. On the final day of the study, they were asked to submit these notes. To be included in the analysis, they had to attend at least nine out of the 12 sessions. Cognitive and physical functioning was measured before the initiation (pre-test) and after completion (post-test) of the program.

### 2.5. Nordic Walking Training (Walking Group)

NW involves walking while holding Nordic ski poles in both hands, allowing older adults who have an unsteady gait to walk with a stable posture. NW has two approaches: an aggressive style, where the upper body leans forward, and a defensive style, where the body is upright [50]. The defensive style was developed in Japan and is also called the Japanese style. In particular, the poles are held straight, with elbows bent at 90°, and placed on the ground in front of the feet. This study used the defensive style because it is more reliable for older adults with orthopedic problems of the spine, hip, and knee. 

The NW group underwent a 60-min training session in pole height adjustment and NW techniques for effective walking. After the participants had reliably mastered the walking technique using the poles, they began training at home. NW was performed at the participant’s pace, three times per week for 4 weeks and they were not instructed to increase the intensity of their walking. Training comprised of 30 min outdoor walking, with an initial warm-up and a final cool-down indoor session (demonstrated on a DVD, each lasting 15 min combined). The total training duration was 45 min. A tutorial DVD was also prepared for cases where a participant might have forgotten how to perform NW. Participants could take breaks as needed during training. 

Participants recorded their training progress on a weekly basis. Fidelity checks were completed during the post-test to ensure that proper procedures and protocols were followed including video recording.

### 2.6. Dance Program Training (Dance Group)

The dance program used in this study was designed by a researcher who was knowledgeable in cognitive and physical functioning and professional dancers with >30 years of experience in choreographing large-scale dance works. Among the 90 participants in our study, 35.56% had dance experience. Of those, 10% had participated in the disco boom that occurred approximately 40 years ago, and 1.11% were part of their high-school dance club. Moreover, as adults, 10%, 2.22%, and 1.11% started ballroom, jazz, and hula dancing, respectively, while 10% started aerobic dance at the gym for health reasons. Only 1.11% of the participants had practiced ballet since childhood. Therefore, we choreographed dance routines using well-known nostalgic songs (released between 1952 and 1983) that would be highly relatable to the participants and have a reminiscence effect [51]. Each soundtrack consisted of four connected songs. The speed of the music was chosen to be suitable for aerobic exercise [52], with songs ranging from 120 to 125 beats per min. Moreover, given the relationship between the shoulder joint flexion angle and cognitive function [46], participants were required to perform a lot of shoulder elevation as part of the upper limb large movements. Various steps using the lower extremities were used to increase the cognitive load as a dual-task exercise [43]. As these steps involved shifting the center-of-gravity, we expected a fall-prevention effect. 

The dance program was prepared by professional dancers and recorded on DVDs. The dance routine on each of the four DVDs (one per week during the 4-week intervention period) was 30 min long. To familiarize themselves with dance training, the dance program group received a 60-min personal training session on dance techniques using a DVD, after which they started training at home. The dance program training was conducted three times per week at the participants’ pace for 4 weeks. The training was conducted indoors, each combining 30 min of the dance with an initial warm-up and final cool-down session lasting 15 min combined, for a total training duration of 45 min, similar to NW training. Although the dance routines could be performed in a sitting position, a standing position was encouraged to improve balance. The participants could take breaks as needed during training. They recorded their training progress every week. Fidelity checks were completed during the post-test to ensure that proper procedures and protocols were followed including video recording.

### 2.7. Control Group

The protein-only group served as the control group and did not undergo any training intervention. These participants were requested to continue their usual routine as much as possible during the 4-week intervention period and to consume the provided protein supplement (details below). They were told that they could participate in the dance program after the study completion.

### 2.8. Protein Intake

In this study, participants from all three groups received protein supplementation three times per week for the 4-week intervention period. Given the reduced ability to synthesize muscle with age [45,53] and reports of intervention trials where muscle wasting occurred upon exercise [46], protein intake becomes essential when older adults perform exercise. Therefore, a protein supplement containing approximately 8.0 g of protein (branched chain amino acids) in a baked cake was provided to all participants. The main ingredients were sesame seeds and dried tofu. The amino acid distribution is presented in Figure 2. To ensure that the protein did not alter their habitual dietary intake, we requested that the participants in the training and control groups consume it before and after training and at 10:00 or 15:00, respectively. All participants recorded their food intake every week. Fidelity checks were completed during the post-test to ensure that the proper procedures and protocols were followed.

### 2.9. Cognitive Function Measures

The MMSE-J was used to assess the inclusion and exclusion criteria and screen participants (cutoff score of 23). We measured the cognitive function using the Montreal Cognitive Assessment (MoCA) [54] and Frontal Assessment Battery at bedside (FAB) [55] before the intervention (Table 1). The questionnaires were provided at an in-person interview. 

The MoCA is a 30-item cognitive screening tool with high sensitivity and specificity for detecting MCI within the normal range of the MMSE. Specifically, one point is added to account for individuals with an educational history of ≤12 years. The total score ranges from 0 to 30 points, with a cutoff of 26, as scores ≤ 26 points indicate global cognitive dysfunction. In this study, the MoCA was used to measure the global cognitive function.

The FAB test provides a score ranging from 0 to 18 points and assesses the executive function. The FAB includes six neurophysiological tasks: similarities (conceptualization), lexical fluency (mental flexibility), motor series (programming), conflicting instructions (sensitivity to interference), go–no-go (inhibitory control), and prehension behavior (environmental autonomy). A lower FAB score indicates a greater degree of executive dysfunction.

### 2.10. Imitation Ability Measures

Dance training focuses on perfecting movements using executive functions and behavioral imitation [41,42]. Therefore, the dance training intervention group was directly trained in imitation ability. A simple hand gesture imitation test can assess visuomotor deficits in individuals with AD with high sensitivity and assess difficulties in those with MCI [56,57]. For the imitation test, we gave imitation commands with the upper limb from the meaningless gesture item of the Standard Performance Test for Apraxia [58] during an in-person interview. 

For this test, the examiner sat facing the participant, instructed them to imitate the hand gestures, which were carefully observed. The one-handed imitation tasks were: (1) Luria’s chin hand test (horizontal), which was imitated with the right and left hands; (2) the fox imitation task (forming a ring with fingers I, III, and IV), which was also imitated with both hands; and (3) forming a ring with fingers I and V, transfer from left to right and then, transfer from right to left. In each case, the examiner presented the gesture until the participant could imitate it, and the maximum observation time for a response was 10 s. The two-handed imitation tasks were (1) intertwining rings with fingers I and II of both hands; (2) crossing both palms, extending fingers II–V upward, and intertwining both fingers I; and (3) simultaneously tapping the desk with one hand in the form of a rock and the other hand in the form of a paper, then, simultaneously exchanging hand shapes and tapping the desk. This was performed three times in a row, switching once every 2 s. The examiner presented the gesture until the participant could imitate it, and the maximum observation time for a response was 15 s. Finally, the continuous one-handed movements were the Luria flexion ring and extension fist task, which required forming a ring with fingers I and II in front of the chest, followed by three repetitions of forming a fist simultaneously with upper limb extension. In each case, the examiner presented a gesture until the participant could imitate it; the maximum observation time for a response was 20 s. One point was added for appropriate correction after an error in all of the tasks, and two points for incomplete imitation. Thus, for perfect imitation ability, the score was 0, whereas, for maximum imperfect ability, the score was 22 points.

### 2.11. Gait Ability Measures

NW is a physical activity recommended for older adults and positively impacts the gait parameters [30]. Therefore, the participants in the NW intervention group were directly trained in walking ability. A faster gait speed is associated with maximal locomotor efficiency while the maximum gait speed is associated with better cognitive function than normal gait speed [59]. Therefore, to assess gait ability, we requested that the participants walk 10 m at a maximum speed (the 10-m walk test) on an indoor flat floor with no nearby obstacles. 

For this test, the central 10 m comprised the test distance with a 1-m flying zone at both ends for acceleration and deceleration. Thus, participants were given a flying start and finish and were instructed to walk the distance back and forth at an even speed and turn at the end mark. At the start of the test, participants were instructed as follows: “Walk as fast as possible, be safe, and do not run.” All participants wore their own comfortable clothing and shoes during the tests. Gait ability was measured using the AYUMI EYE gait analysis device (Waseda Elderly Health Association Co., Ltd., Tokyo, Japan), which measures the walking speed and analyzes gait using a single-point acceleration sensor. The device weighed 18.5 g (including the battery), was 62.4 × 30.9 × 11.8 mm in size, and was attached using a rubber belt to align with the third lumbar vertebra of the spine when the participant was in a standing posture. An iOS application (Apple Corp., Cuppertino, CA, USA) designed for the module received the gait-related parameters including the walking speed, stride length, root mean square (RMS), and gait cycle variability from the device. 

The walking speed (m/s) was calculated using the time taken to cover 10 m, and the stride length (cm) was determined by the distance covered divided by the number of steps. RMS (1/m) represents the degree of left-right swaying during walking, and thus, the smaller the value, the better the balance. This was obtained by dividing the left and right acceleration by the square of the walking speed. The variability of the gait cycle is represented by the standard deviation of the time taken for one gait cycle (s). One gait cycle in this device refers to the time from left heel contact to right heel contact and back to left heel contact. A smaller variance of the gait cycle results in a more stable gait rhythm. We measured the walking speed in both directions and used the faster direction for the analysis.

### 2.12. Mood State Measures

Several psychological measures were assessed before and after the intervention. We used the Geriatric Depression Scale (GDS-15) [60] and the World Health Organization Quality of Life-26 (WHOQOL-26) questionnaire [61] to assess the quality of life (QOL) (Table 1). 

The GDS-15 Japanese version [62] is a 15-item questionnaire that interviewers can use to assess the symptoms of depression in older adults. The responses are provided in a “yes or no” format to be easily understood by older people who have impaired cognitive function. The total score ranges from 0 to 15 points, with a cutoff of 5 points. A higher score reflects more depressive symptoms. 

The WHOQOL-26 is a self-administered questionnaire that assesses the QOL comprising 26 items in five domains: the physical, psychological, social, environmental, and overall QOL. The participants rated each item on a 5-point scale (i.e., very poor, poor, undecided, good, and very good). The mean score for each domain was calculated, with a higher score indicating a better QOL.

### 2.13. Physical Function Measures

Using the Short Physical Performance Battery as a reference [63], we conducted a standing balance test and the Five Times Sit-to-Stand Test (FTSST). For the standing balance test, the participants performed side-by-side, semi-tandem, and tandem balance tests, each for 10 s. For the FTSST, the participants sat with their backs against a straight-backed chair, folded their arms in front of their chest, and performed five chair stands as quickly as possible. A stopwatch was used to measure the time from the starting cue to when the participants’ buttocks touched the seat after the fifth rise [64].

Grip strength (kg) was measured using a digital dynamometer (Takei D T.K.K.5401, Takei Scientific Instruments, Tokyo, Japan). Two maximum force trials were performed with each hand and the highest measurement was used for the analysis.

Toe grip strength (kg) was measured using a digital toe-grip dynamometer (Takei D T.K.K.3362, Takei Scientific Instruments). The participants sat on a chair barefoot with their hips and knees flexed at approximately 90°. The ankles were fixed in a neutral position using a strap. The first phalanges were placed at the grip bar, and a heel stopper was adjusted to fit the heel. Then, they gripped the bar with the maximum contraction of the toe flexors. Two maximum force trials were performed with each foot, and the highest measurement was used for the analysis.

The single-leg stance test was performed on a flat surface, and the elapsed time was measured using a stopwatch. The participants balanced on one leg at a time, without shoes, and with their eyes open for 120 s. They could alternate between the two legs, resting as needed. If they lost their balance and stood on both feet before the 120 s had elapsed, the time of single-leg stance was recorded. Two trials were performed on each side, and the best time was used for the analysis.

The dance program included choreography requiring shoulder flexion as a major movement in the upper extremity. Therefore, to assess the improvement in upper limb elevation, the active range of motion of shoulder flexion of the dominant upper limb was performed. A 300-mm University of Tokyo type stainless steel goniometer (Frigz Medico Japan Co. Ltd., Chiba, Japan) was used for this measurement, during which the participant was in a sitting position while one physiotherapist checked the joint position, and another physiotherapist measured the participant’s active flexion angle.

Balance function for postural maintenance can be achieved through an ankle strategy, a hip strategy, or a combination of both [65], but the correlation between the ankle muscle strength and postural control becomes affected with age [66]. The functional capacity of the ankle plantar flexors is important for balance tasks [67]; in contrast, the functional capacity of the ankle dorsiflexors is related to the tendency to fall [68]. Therefore, to assess the neuromuscular function and the performance of the ankle joint in older adults, we measured the angle of the heel lift and toe lift to determine the ankle plantar flexor function and ankle dorsiflexor function, respectively. For the heel lift test, the participants stood barefoot against a wall with their hands outstretched at shoulder height for balance, elbows slightly bent, and feet hip-width apart. At the start signal, they raised their heels as high as possible for 5 s with their knees fully extended. The maximum height at which the participant’s heels were raised from the ground was measured using a ruler (*b*). The distance from the participant’s first metatarsophalangeal joint to the heel was also measured (*a*), and the angle of the heel lift was calculated using $\theta =Atan\left(b/a\right)$.

For the toe lift test, which was performed after the heel raise test, the participants positioned themselves in the same way as the heel raise test, placing their feet flat on the floor. They were instructed to raise their toes as high as possible while keeping only the heel of their feet on the floor for 5 s at the start signal. The highest toe raise from the ground was measured using a ruler (*c*). The angle of the toe raise was calculated as $\theta =Atan\left(c/a\right)$.

### 2.14. Body Composition Measures

Dual-energy X-ray absorptiometry is a well-established conventional method for assessing body composition [69,70]. More recently, bioelectrical impedance analysis (BIA) has been used because of its similarity to dual-energy X-ray absorptiometry [46,71]. BIA can record reliable and noninvasive measurements by passing a weak electric current through the body. Given the ease of measuring different body types, it has been used in large epidemiological studies nationwide and in clinical settings [72]. We used the InBody S10 (Biospace Co., Ltd., Seoul, Korea) device for BIA to measure the basic body composition parameters: muscle mass (kg), fat mass (kg), body mass index (kg/m^2^), and the skeletal muscle mass index (SMI) (kg/m^2^).

In older adults, muscle degradation exceeds muscle synthesis, making it difficult to increase muscle mass and improve other body composition indices. However, even in the absence of an increase in muscle mass, changes in muscle quality due to muscle overhydration can be confirmed using extracellular water/total body water analysis (ECW/TBW). The phase angle (PhA) is the resistance (reactance, *Xc*) measured when an alternating current passes through the cell membrane and is expressed as an angle. As the condition of the cell membranes is assessed based on the conductivity of the tissue, higher PhA values indicate better nutrition and health [73]. Recent studies have reported that PhA is associated with muscle quality, inflammatory and oxidative stress biomarkers, physical function in older adults, and survival [74,75,76,77]. Therefore, ECW/TBW and PhA can be used to identify changes in muscle quality, even when there is no gain in muscle mass. For the whole-body PhA, the BIA-measured impedance (*Z*) and *Xc* of the right arm, trunk, and right leg at 50 kHz were summed and calculated as PhA: $\theta =Arcsin\left(Xc/Z\right)$. Measurements were performed with the participant sitting in a chair; eight electrodes were attached (one on the thumb and middle finger of each hand, and one on both sides of each ankle), moistened using an electrolyte-wetting agent, and electrode holders were connected using the four-electrode method.

### 2.15. Other, Frailty, and Sarcopenia Measures

Background information such as age, sex, education (< or >12 years), height (cm), weight (kg), work status, income style, number of family members, exercise habits, and distance from the nearest railway station to home was collected in an interview. Medical history data (i.e., diabetes, heart failure, hypertension, angina pectoris, myocardial infarction, stroke, Parkinson’s disease, chronic lung disease, depression, dementia, atrial fibrillation, hearing and visual impairment, prosthesis, arthritis, and limb surgery) was also assessed in an interview.

To assess the activities of daily living (ADL), the Barthel index [78] and Instrumental Activities of Daily Living scale (IADL) were used. For the IADL, we used the Tokyo Metropolitan Institute of Gerontology Index of Independence in Daily Living [79] to assess higher-order life functions, given that the participants were healthy community-dwelling older adults. This index evaluates the IADL and the subscales of intellectual activity and social role.

In this study, we used the unintentional weight loss of ≥5% over the past 2 years and a response of “No” to the GDS-15 sub-item “Do you feel that you are full of vitality?” [80].

The inability to perform the FTSST, grip strength, and calf circumference (cm) were used as indicators of sarcopenia [81]. We also used the SMI (kg/m^2^), which is a measure of muscle mass and sarcopenia [82].

### 2.16. Statistical Analyses

All participants were included in the study based on an intention-to-treat analysis. Baseline equivalence between the three groups was examined using the analysis of variance. To identify differences between the three groups, the change in scores (post-intervention minus pre-intervention) was calculated for the following variables: cognitive function, gait, imitation, mood state, physical function, and body composition.

For the missing data, we employed multiple imputation, which is a single assignment method repeated *m* (>1) times to avoid bias due to missing data. The estimates calculated for each pseudo-complete data are integrated as missing values. The inductive model included the age, sex, pre-intervention score, post-intervention score, and score changes as characteristic variables. Using the Multivariate Imputation by Chained Equations algorithm [83], we performed a multiple assignment method by chaining equations. The number of pseudo-complete data, *m*, was set to 500 to ensure accurate estimation and testing [84].

An analysis of covariance with permutation tests (ANCOVA) was performed for each change in score between the three groups because it is applicable for analyzing small sample sizes and can correct for false positives. The respective baseline scores for cognitive function, imitation ability, and psychological measures were used together with the sex and age as covariates. Additionally, for the physical function and body composition measures at the baseline, sex, age, height, and weight were included as covariates because these measures affect the body size. All ANCOVAs were subjected to permutation tests using the “aovp” function of the lmPerm package [85]. The permutation test is significant when the sample size is limited [86], making it suitable for testing the effects of intervention trials with small sample sizes [87,88,89]. We subsequently adjusted all *p*-values using Storey’s false discovery rate correction method (Table 2) [90]. If the f-ratio value between the adjusted mean scores of the three groups was significant, the Scheffé test was used as a post-test to determine the difference between each pair of groups. Furthermore, the effect size (*η*^2^) was calculated as the sum of squares between the groups and the sum of squares from the ANCOVA permutation test using eta squared [91].

Statistical significance was set at a two-sided *p* < 0.05, and all analyses were performed using R version 4.0.3 (R Foundation for Statistical Computing, Vienna, Austria).

## 3. Results

### 3.1. Background Characteristics

Among the final 88 participants (mean age, 67.81 years; SD, 5.64; range, 60–82 years), no one had ADL issues, 29.55% were women, and 47.73% had full- or part-time jobs. The IADL score was 100%. Moreover, 94.31% had a perfect score for intellectual ADL, while 5.68% did not read the newspaper. For social ADL, 31.82% had a perfect score; the rest had a low score because they refrained from social activities due to COVID-19. Furthermore, 56.82% of the participants had no exercise habits, while 18.18% performed regular exercise and 25.00% had exercise habits including competition. Participants had a mean education history of 15.58 years (SD, 1.5663). Their MMSE-J scores were above the cutoff of 23 points, with a mean score of 28.20 (SD, 1.45; range, 25.00–30.00) points. The mean MoCA score was 26.80 (SD, 1.76; range, 21.00–30.00) points, which was also above the cutoff of 25 points.

All participants could perform the three standing balance tests (side-by-side, semi-tandem, and tandem) for 10 s and the FTSST. Regarding frailty, 2.27% of the participants mentioned unintentional weight loss of ≥5% over the past 2 years. For the GDS-15 sub-item, “Do you feel that you are full of vitality?”, 25% answered “no.” Therefore, no participants were classified as frail for more than two items, as per the Study of Osteoporotic Fractures Index [92] and 27.27% of participants were classified as pre-frail (classified as frail for one item).

The mean calf circumference among women was 34.27 cm (SD, 2.12; range, 29.80–37.60 cm), which is above the Asian Working Group for Sarcopenia (AWGS) cutoff of 33 cm for sarcopenia. Even for men, the mean calf circumference exceeded the AWGS cutoff of 34 cm, with a mean of 36.32 cm (SD, 2.65; range, 30–43.25 cm). The mean maximum grip strength (20.72 kg) in women exceeded the AWGS cutoff of 18 kg for sarcopenia (SD, 3.29; range, 14.40–26.85 kg). Similarly, the mean maximum grip strength for men (31.18 kg) exceeded the AWGS cutoff of 28 kg (SD, 8.31; range, 6.60–55.2 kg). The mean SMI for women was 6.03 kg/m^2^ (SD, 0.40; range, 5.2–6.7 kg/m^2^), which is above the AWGS cutoff for sarcopenia (5.8 kg/m^2^). Men also exceeded the AWGS sarcopenia cutoff of 7.0 kg/m^2^, with a mean of 7.65 kg/m^2^ (SD, 0.76; range, 5.8–10.1 kg/m^2^). One participant had a hip prosthesis and five had an anterior cruciate ligament (ACL) injury, all of whom were >10 years post-treatment and had a good prognosis. Upon physical examination, 43.18% of participants had osteoarthritis of the knee (27 in one leg and 11 in both legs).

After informed consent was obtained, 29, 30, and 29 participants were randomly assigned to the walking, dance, and control groups, respectively. Their baseline measurements are shown in Table 1.

During the 4-week period (12 training sessions), 86 out of the 88 participants completed all tests and measurements and at least 11 training sessions. However, two individuals from the walking group discontinued participation; one underwent surgery and the other refused participation. Therefore, data from 27, 30, and 29 participants in the walking, dance, and control groups, respectively, were included in our analysis (Figure 1).

### 3.2. Cognitive Function

The pre-intervention cognitive function scores are presented in Table 1, and the changes in scores for the MoCA and FAB are presented in Table 2.

We found significant differences in the MoCA total score between the three groups. The dance group had a significantly larger change in the mean MoCA total score than the control (*p* = 0.0000; 95% confidence interval [CI] = 1.2007–3.4154) and walking groups (*p* = 0.0135; 95% CI = −2.4910–−0.2350), whereas there was no difference in the mean score between the walking and control groups (*p* = 0.1233; 95% CI = −0.1921–2.0823). Furthermore, there was a significant difference in the total FAB score between the three groups. Compared to the control group, the dance (*p* = 0.0006; 95% CI = 0.5022–2.1369) and walking groups (*p* = 0.0369; 95% CI = 0.0432–1.7218) had a significantly larger mean change in the FAB total score, whereas the difference in the mean score between the walking and dance groups was not significant (*p* = 0.4282; 95% CI = −1.2696–0.3955).

We observed significant differences between the groups for the visuospatial/executive MoCA sub-score, where the dance group had a significantly larger mean score change than the control (*p* = 0.0050; 95% CI = 0.1920–1.2885) and walking groups (*p* = 0.0296; 95% CI = −1.1659–−0.0490), but there was no difference between the walking and control groups (*p* = 0.8413; 95% CI = −0.4302–0.6958). Moreover, for the language sub-score, a significant difference was found between the three groups with the walking group having a significantly larger mean score change than the control group (*p* = 0.0271; 95% CI = 0.0458–0.9478), but the difference between the dance and control groups (*p* = 0.4411; 95% CI = −0.2128–0.6656) or between the walking and dance groups (*p* = 0.3262; 95% CI = −0.1770–0.7178) was not significant. Additionally, for the orientation sub-score, there was a significant difference in the improvement between the three groups; compared to the control group, the walking (*p* = 0.0256; 95% CI = 0.0309–0.5897) and dance groups (*p* = 0.0038; 95% CI = 0.1049–0.6491) had a significantly larger mean score change, whereas there was no difference between the walking and dance groups (*p* = 0.8356; 95% CI = −0.3438–0.2105). Finally, there were no significant differences in the naming, attention, abstraction, or delayed recall sub-scores of MoCA between the three groups.

In the FAB test, we observed significant differences between the three groups for the go–no-go (inhibitory control) sub-score. Compared to the control group, the walking (*p* = 0.0060; 95% CI = 0.1736–1.2466) and dance groups (*p* = 0.0017; 95% CI = 0.2580–1.3029) showed a significantly larger mean score change, but there was no difference between the walking and dance groups (*p* = 0.9471; 95% CI = −0.6025–0.4618). However, there were no significant differences between the three groups in any of the other FAB sub-scores including similarities (conceptualization), lexical fluency (mental flexibility), motor series (programming), conflicting instructions (sensitivity to interference), or prehension behavior (environmental autonomy).

### 3.3. Mood State

There was no significant difference in the WHOQOL-26 or GDS-15 scores between the three groups before and after intervention (Table 2).

### 3.4. Imitation Ability

We found significant differences in the total score of the gesture-based imitation task between the three groups before and after intervention (Table 2). There were significantly fewer errors in the total score of imitation in the dance group compared to the walking group (*p* = 0.0005; 95% CI = 0.85467–3.5231), whereas there was no difference between the dance and control groups (*p* = 0.0847; 95% CI = −2.4948–0.1247) or between the walking and control groups (*p* = 0.1834, 95% CI = −0.3412–2.3489).

### 3.5. Gait Ability

We observed significant differences in the maximum gait speed over 10 m between the three groups before and after intervention (Table 2). The dance group presented a significantly faster gait speed compared to the walking group (*p* = 0.0038; 95% CI = 0.1457–0.8991), whereas no significant differences were found between the walking and control groups (*p* = 0.1820; 95% CI = −0.0958–0.6637) or between the dance and control groups (*p* = 0.2798; 95% CI = −0.6082–0.1313).

There were no significant differences in the stride length, RMS, or gait cycle between the three groups.

### 3.6. Physical Function

For the angle of the heel lift (calf raise test), there was a significant difference between the three groups. Compared to the control group, the walking (*p* = 0.0083; 95% CI = 0.4832–3.9343) and dance groups (*p* = 0.0008; 95% CI = 0.9963–4.3569) had a significantly larger angle, but the difference between these groups was not significant (*p* = 0.7930, 95% CI = −2.1795–1.2438). Moreover, for the single-leg stance test, the walking group showed a significantly increased time compared to the control group (*p* = 0.0186; 95% CI = 2.2494–30.1941). However, there was no difference between the dance and control groups (*p* = 0.5892; 95% CI = −7.9775–19.2341) and between the walking and dance groups (*p* = 0.1690; 95% CI = −3.2662–24.4531) in the single-leg stance test.

Concerning muscle strength, there was no significant difference between the three groups in the dominant hand grip strength, toe strength, FTSST, angle of active flexion of the dominant shoulder in the upper limb, or the toe raise test. 

### 3.7. Body Composition

Regarding the body composition parameters, there was a significant difference between the groups in the ECW/TBW, with the dance group having a significantly smaller mean change in the ECW/TBW compared to the control group (*p* = 0.0356; 95% CI = −0.0034–−0.0001) (Figure 3). However, no significant differences were found between the walking and control groups (*p* = 0.0859; 95% CI = −0.0032–0.0002) or between the walking and dance groups (*p* = 0.9498; 95% CI = −0.0015–0.0019). Furthermore, there were no significant differences between the three groups in weight, muscle mass, fat mass, or PhA.

## 4. Discussion

We investigated the effects of two different exercise programs, NW and an original dance program, performed three times per week for 4 weeks, on the cognitive and physical functions of healthy older adults during the COVID-19 pandemic.

### 4.1. Cognitive Function

The total scores for the MoCA and FAB cognitive function measures were improved after performing both exercise interventions. Particularly, the dance program improved the total MMSE-J score, the primary outcome measure of global cognitive function, compared to both the walking and control groups. Furthermore, walking and dance exercises improved the total FAB score for executive function compared to no exercise. Specifically, the go–no-go (inhibitory control) FAB sub-score was improved due to walking and dance compared to no intervention. The effects of the two types of exercises on improvement were different, where both NW and dance improved the executive function, but only dance improved the general cognitive function. Walking did not affect the total MoCA score or the visuospatial/executive sub-score, whereas dance improved these compared to both the walking and no intervention. Walking led to an improvement in the language sub-score, while for the orientation sub-score, both walking and dance led to an improvement compared to no intervention. 

Interestingly, the dance group showed a larger improvement in cognitive function than the other two groups. This group demonstrated an improvement in the total MoCA score. This improvement was significantly larger than that observed after NW. Although we used an original dance program in this study, it improved the global cognitive function, similar to a recent meta-analysis [93].

Older adults at a high risk of falling cannot maintain a conversation while walking, as evidenced by the “stops-walking-when-talking” phenomenon [94]. When performing two or more tasks simultaneously, cognitive resources are divided to accommodate for the processing needs. However, older adults become overloaded with information processing, making it difficult to perform simultaneous tasks due to a lack of resources [95]. Dance improves dual-task performance in older adults with variability in gait and cognitive tasks. It is a concurrent dual-task training where the steps and movements are memorized, planned, and cognitively processed while performing aerobic exercise [43]. Thus, this activity improves the global cognitive function, and this effect is specific to dance.

The total FAB score, which assessed the executive function, a cognitive domain vulnerable to age-related changes in brain function and structure, was improved after NW and dance training. A recent meta-analysis reported that aerobic exercise, among other types of physical exercises, improved executive function in healthy older adults [96]. Therefore, our results may also be attributed to the effects of aerobic exercise. Regarding the effects of dance, some previous studies have reported improvements in executive function and neuroplasticity [35], while others have found little or no significant difference [93]. In particular, a previous study reported that ballroom dancing did not improve executive function compared to walking intervention [97]. However, these studies included healthy older adults who had no cognitive impairment. There are two reasons for the discrepancy between our results and those of the above-mentioned studies. First, we could not measure the participants’ physical activity because of the COVID-19 pandemic, although our dance program was created with an aerobic effect in mind. Hence, it is possible that the exercise load was greater than in previous studies. Even the speed of the music we used for the whole dance routine was approximately 120 beats/min, which is faster than the spontaneous motor tempo [98] and is probably at the speed of the dance forms that were used in one of the studies reviewed by Nascimento [35]. Hence, we probably achieved an aerobic effect. Second, similar to previous studies [35,93,96,97], our participants were healthy older adults with no ADL, IADL, or cognitive function issues. However, this study commenced in November 2020 during the ongoing COVID-19 pandemic, just before the peak of the third wave [99]. The level of physical activity of older adults decreased in the 3 months before (January 2020) and during (April 2020) the first wave of the COVID-19 pandemic in Japan [100]. Therefore, there may have been room for improvement due to the inherent cognitive decline related to social restrictions.

We also observed an improvement in the visuospatial/executive sub-score of the MoCA after dance training. A recent meta-analysis also found that visuospatial skills were improved with dance, and this has been observed for neural mechanisms [101], in line with our results. In another study comparing dance and fitness training, both improved visuospatial skills [40]. In our study, the task content of the visuospatial/executive sub-score was related to executive functions including alternating Trail Making and Clock Drawing tests, and the NW group did not show any improvements in these. The correct posture for NW requires looking and moving forward, which may have contributed to the lack of spatial cognition training. 

We observed an improvement in the language sub-score of the MoCA after NW training, but not dance training. In a previous meta-analysis, dance was found to be more effective for improving language fluency than Tai Chi or yoga [102]. Therefore, a high exercise load may be important for improving the effect of physical activity on language. Here, our dance program prioritized the constant learning of new choreography, and thus may not have been as effective as NW, which constantly added exercise load. 

We also observed improvements in the MoCA orientation sub-score after NW and dance training. Orientation includes multiple dimensions of cognitive function and is susceptible to MCI and dementia. In a large-cohort prospective study conducted among older adults, the Six-item Cognitive Impairment Test score, which includes many items related to orientation, was significantly worse in the physically inactive group than in the moderately- or highly-active group [103]. Therefore, in our study, NW and dance training improved this parameter due to physical activity. In addition, social engagement can also contribute to cognitive improvement [104,105]. However, all the participants in this study (including the control group) were requested to keep a daily training and protein intake diary (three times per week after training). Therefore, we believe that social engagement did not differ among groups and that the orientation improvement was attributed to the physical activity.

Furthermore, we observed an improvement in the go–no-go (inhibitory control) sub-score of the FAB after both NW and dance training. The inhibitory control task elicits false alarm motor responses following the inhibition of inappropriate responses [55]. Response inhibition is necessary for high-level executive control by moving the body appropriately to precisely-timed music. Music training has domain-independent transfer effects [106] and may enhance inhibitory control abilities [107]. However, we found no significant difference between the participants in the dance and NW groups that trained with and without music, respectively. A recent meta-analysis investigating the effects of exercise interventions on subdomains of executive function in older adults reported that exercise improved inhibitory control [108]. Therefore, NW and dance training improved inhibitory control due to exercise training.

Moreover, in a previous study, an in-person exercise intervention improved executive function in 4 weeks [109]. Our study suggests that an individualized NW and dance program for the same period can also be effective.

### 4.2. Mood State

The exercise interventions did not significantly affect the QOL or the GDS. However, there was a trend in the dance group to improve the mood state measures (WHOQOL-26 index) in healthy older adults. Physical activity in older adults is related to well-being, and aerobic training is the most beneficial [110], leading to both short- and long-term improvements [111]. In previous studies, older adults could assemble and perform exercise interventions, but the present study was conducted in a situation where individuals performed the interventions at home during the COVID-19 pandemic. Our results show that QOL can be improved by exercise intervention without social involvement. A recent meta-analysis reported loneliness and depression due to limited access to physical activity during the COVID-19 pandemic [112]. Therefore, although our participants were healthy older adults, the impact of being under social restrictions may have reduced their QOL, providing more room for improvement. In addition, a previous study that focused on dance to address social isolation during the COVID-19 pandemic [113] demonstrated that dance is efficient with respect to improving QOL.

### 4.3. Imitation Ability

The results of the dance-training intervention, which directly trained imitation ability, indicate that dance improved the imitation ability compared to walking. The imitation gesture total score was significantly improved in the dance group compared to the NW group.

Dementia leads to difficulties in mimicking the commands and gestures required for motor therapy and programs [114,115]. In AD, visuospatial dysfunction is assessed by the ability to imitate hand positions, and performance in the hand imitation test can identify patients with mild AD [57,116]. The hand imitation test is also used as a screening test to investigate the severity of the disease [117]. Previous studies on dance interventions in older adults have not validated or reported changes in imitation ability, a direct training component of dance. However, dance interventions have increased visuospatial functional tasks [97] and the brain cortex related to imitation [42]. Here, the dance group demonstrated significantly improved imitation scores and visuospatial/executive skills required for imitation. The NW group did not require visuospatial/executive skills and had significantly worse imitation scores. Thus, the effects of the different interventions on cognitive function are reflected in these results. The hippocampus is important for visuospatial cognition. Interestingly, a previous study comparing 18 months of dance and repetitive motion intervention, very similar to the current study, found that both groups had an increased hippocampal volume, and there was an increase in the hippocampal subfield area in the dance group [39]. Dance has an advantage concerning cognitive improvement, as there are no studies reporting that dual-task training is inferior to single-task training for cognitive transmission. Even in the absence of aerobic exercise, juggling, which requires visuospatial function and sensory movement, has been shown to increase hippocampal volume in older adults [118]. In this study, the original dance program required learning new choreography and would have been less strenuous. However, it still produced an aerobic effect comparable to NW and an improvement in cognitive function, which is vulnerable in older adults because of the mimetic training effect.

### 4.4. Gait Ability

Dance training improved gait parameters more than NW training despite the greater direct gait training aspect of NW. Specifically, dance training led to a larger improvement at the maximum walking speed over 10 m compared to the NW training. In previous studies, normal walking speed, measured as the gait parameter, was improved in the NW [119] and dance groups [31]. Walking speed is related to executive function in older adults [120,121], and the maximum walking speed is more strongly correlated with cognitive function than normal walking speed [59]. Therefore, as the participants in this study were healthy older adults, we assessed the maximum walking speed over 10 m instead of the normal walking speed as the gait parameter. Moreover, although NW training requires the correct walking technique, the participants were not trained to walk fast. Interestingly, the NW group showed significant improvement in the walking cycles related to the timing discrepancy between the left and right balance. Moreover, the duration of standing on a single leg was significantly longer in the NW group. Although many previous studies have reported improvements in balance after dance intervention [31,39,122,123,124], our results did not show that such changes occurred after dance training. In the NW and dance groups, the angle of the heel lift was significantly improved, but there was no change in the toe lift. The heel lift reflects the plantar flexion of the ankle joint in the standing position, and the plantar flexors required for this movement constitute one of the major postural muscles [125], which are also important for propulsion during running [126]. Plantar flexors are associated with balance and walking speed in older adults [67]. NW training did not improve propulsion ability (walking speed), but it improved balance (duration of standing on a single leg). Conversely, dance training did not improve balance, but it improved propulsion ability. Conventionally, NW includes a significantly greater strength training element than normal walking [127] and is more effective for extensor muscles such as the gluteus maximus and hamstrings, which are important for propulsion. In this study, the stride length for NW training might have been too small to improve the propulsion ability. Additionally, although the dance program included many steps requiring center-of-gravity shifts, it did not lead to an improvement in balance because the participants did not execute the change in center-of-gravity as required. Similarly, the shoulder flexion angle did not improve in either the NW group, which required repetitive shoulder elevation, or the dance group, which required a large amount of shoulder elevation. 

There were no significant differences in muscle strength concerning changes in the grip (upper limbs) or toe strength (lower limbs). Therefore, the month-long aerobic exercise intervention program did not change the muscle output, but may have changed the performance of neurological bodily functions such as the heel lift, with dance training being particularly efficient in eliciting these changes.

### 4.5. Body Composition

Dance training led to a reduction in ECW/TBW compared to NW and no intervention. However, there was no change in the muscle composition in the NW or dance group compared to the control group. 

In our previous study, we found that older adults with dementia living in nursing homes had reduced limb measurements, SMI, and total body protein levels. Furthermore, muscle deterioration increased, and muscle synthesis decreased in response to exercise load [46]. In this work, the mean sarcopenia assessment index was above the cutoff value for healthy older adults. However, given the poor-dietary quality of this population while practicing social distancing [47], we aimed to avoid a decrease in muscle composition because of exercise load owing to a reduced myosin-synthesis response. Thus, we instructed all participants to consume supplemental food equivalent to 8 g of amino acids three times per week for the duration of the intervention (4 weeks), referring to previous studies [128]. Given that there were no participants who did not consume supplemental amino acids in this study, the effect of amino acid intake is unknown. However, it should be noted that the index related to muscle composition did not deteriorate in the exercise intervention groups. Furthermore, we investigated the relationship between the PhA and ECW/TBW to assess the muscle quality [129]. PhA, which represents muscle mass and cell nutritional status, showed no significant difference between the exercise groups, but it decreased in the control group. Conversely, the ECW/TBW, which reflects changes in whole-body water balance, was improved more in the dance than in the NW group and deteriorated in the control group. Although we hypothesized that there would be more positive effects on the muscles of the NW participants due to the strength training component, surprisingly, we found that muscle quality was improved in the dance group, which requires more left-to-right movements and complex center-of-gravity shifts compared to the back-and-forth repetitive movements required for NW.

This study was an intervention trial conducted during the COVID-19 pandemic, when the state-of-emergency declaration was temporarily lifted. Older adults were at high risk of contracting COVID-19 and had been adversely affected by social distancing and physical activity restrictions. To maintain or improve the health of older adults at home, we developed an exercise program and delivered it in a way that even older people could perform, despite the digital limitations. All included participants could participate and implement the program until the end of the intervention period. Importantly, the ability to improve the cognitive and neurological physical functions, after only 4 weeks of training, provided an important proposition. In addition, we validated the different types of exercise programs such as indoor versus outdoor, providing a choice to older adults. In Japan, where there is no dance culture, offering dance programs based on music familiar to older adults could be a cooperative tool. Dance is a physical activity that requires a variety of cognitive functions [130].

### 4.6. Limitations

This study had several limitations. First, the intervention period was only 4 weeks, rather than the more common 12- or 16-week exercise intervention. This was the period we set aside for post-evaluation measurements to ensure that we were unaffected by any emergency declarations. Nevertheless, considering the continuation of the COVID-19 pandemic and the inability of indoor dance programs to become affected by the state-of-emergency declarations, longer interventions with follow-up assessments are necessary in the future. Second, this study did not provide feedback on the performance accuracy during the exercise program. A normal exercise program would have a teacher or facilitator intervening to point out any errors and guide accuracy during training. As the participants underwent the exercise program on their own and may not have performed the physical movements as required by the program, they may not have improved certain parameters such as arm raising. We should create a system that can provide feedback on the accuracy of the execution of the exercise. Third, we did not determine the exercise load, which is essential for exercise intervention studies. We could not measure the amount of exercise performed by the participants because it was imperative to prevent the spread of infection. Finally, there was no index to measure the imitation ability for the whole body; thus only the imitation ability of the upper limbs was assessed.

## 5. Conclusions

This study was designed to investigate the effects of a NW and dance intervention on the cognitive and physical functions of healthy older adults following social distancing at home during the COVID-19 pandemic. The participants did not have dementia or sarcopenia. For 1 month, the participants underwent an at-home exercise program. Cognitive function assessment showed that both programs improved the executive function due to aerobic exercise compared to the control group. In addition, the dual tasks of learning choreography and aerobic exercise performed by the dance group had a beneficial effect on the global cognitive function. Given that the imitation ability has been associated with cognitive decline [131] and dementia [114,115], the positive effects of dance on primary imitation training are an important component of successful aging. Furthermore, the participants in the dance group showed greater improvement than those in the NW group who received direct gait training in the maximum gait speed, reflecting the cognitive function well [59]. The muscle mass or muscle output from either intervention groups were not significantly different from those of the control group. However, our results suggest that neurologically, exercise may alter the performance of physical function and improve the angle of heel lift. The fact that dance was accepted by older adults, with neither culture nor the opportunity to experience it in Japan, might make it a powerful tool for creating physical activities that can improve cognitive and physical functions.

## Figures and Tables

**Figure 1 ijerph-19-06202-f001:**
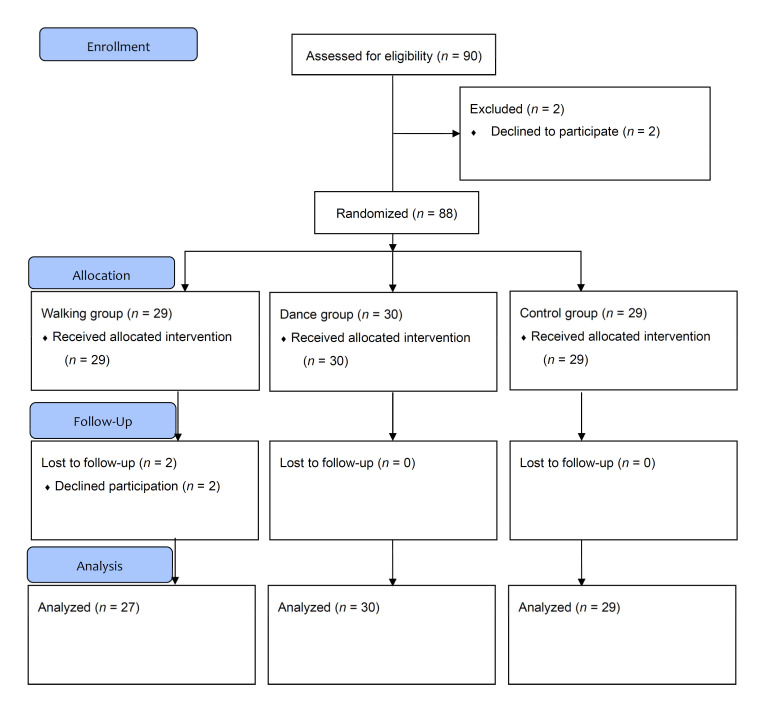
Consolidated Standards of Reporting Trials (CONSORT) diagram.

**Figure 2 ijerph-19-06202-f002:**
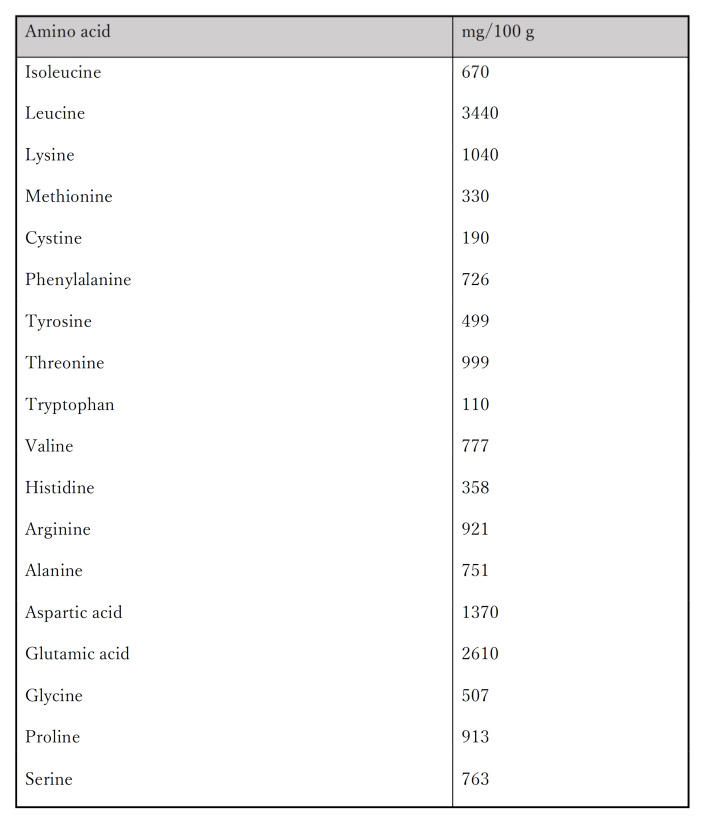
The amino acid analysis of the provided supplemental protein food containing approximately 8 g of amino acids.

**Figure 3 ijerph-19-06202-f003:**
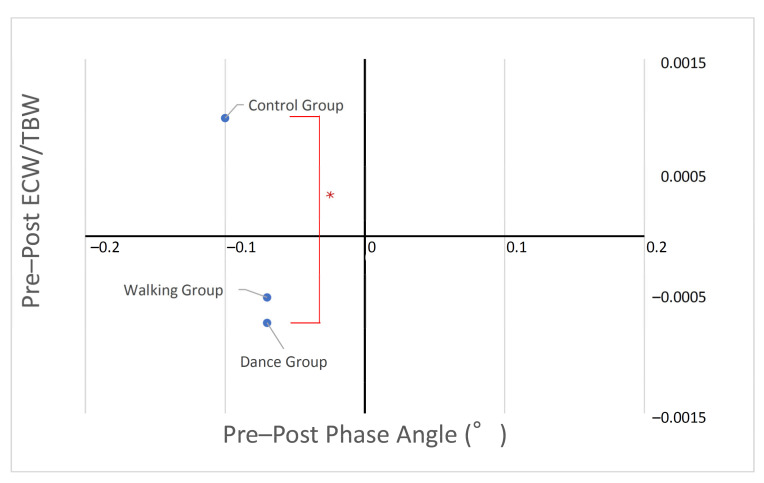
Comparison of the muscle quality by phase angle and extracellular water/total body water analysis based on the amount of change in the three groups. There was a significant difference between the dance and control groups in ECW/TBW at the red line. * *p* < 0.05.

**Table 1 ijerph-19-06202-t001:** The participant characteristics at baseline.

Criterion	Walking Group		Dance Group		Control Group		
	(*n* = 29)		(*n* = 30)		(*n* = 29)		
	Mean (SD)	Range	Mean (SD)	Range	Mean (SD)	Range	ANOVA *p*-Value
Age, years	67.93 (5.81)	60–80	67.20 (5.39)	60–82	68.31 (5.87)	61–79	0.7997
Sex (female: male)	7:22		13:17		6:23		
MMSE total score	28.17 (1.28)	25–30	28.60 (1.33)	26–30	27.83 (1.65)	25–30	0.3674
Education, years	15.93 (1.65)	12–20	15.37 (1.56)	12–18	15.45 (1.48)	10–18	0.2428
Number of family members	2.41 (0.91)	1–4	2.17 (0.83)	1–4	2.00 (0.76)	1–4	0.1707
IADL (13 categories)	11.97 (0.98)	10–13	12.10 (0.88)	10–13	11.90 (0.77)	10–13	0.7666
Calf circumference (cm)	35.84 (3.01)	30.00–43.25	36.01 (2.83)	29.80–42.50	35.28 (2.08)	31.50–40.25	0.4260
Work status (No job:Work part-time:Work full-time)	16:5:8		15:7:8		15:6:8		
Exercise habits (No or Light:Moderate:Heavy)	15:6:8		22:3:5		13:7:9		
Lifestyle diseases (None:With)	16:13		21:9		19:10		
Diabetes (None:With)	27:2		28:2		29:0		
Weight loss	29:0		28:2		29:0		
A felling of fatigue	28:1		27:3		27:2		
Hip Arthroplasty (None:One-Leg:Both-Legs)	29:0:0		30:0:0		28:1:0		
Knee Injury (None:OA of One-Leg:OA of Both-Legs:ACL of One-Leg:ACL of Both-Legs)	17:6:3:3:0		15:8:6:1:0		13:13:2:1:0		
Cognitive Function Measures							
MoCA total score	26.76 (1.68)	23–30	26.63 (1.83)	22–30	27.00 (1.79)	21–30	0.6036
Visuospatial/Executive	4.10 (0.86)	2–5	4.03 (0.93)	2–5	4.28 (0.92)	2–5	0.5736
Naming	2.97 (0.19)	2–3	3.00 (0.00)	3	2.97 (0.19)	2–3	0.5979
Attention	5.86 (0.44)	4–6	5.63 (0.61)	4–6	5.83 (0.38)	5–6	0.1613
Language	1.79 (0.90)	0–3	2.03 (1.13)	0–7	1.79 (0.82)	0–3	0.5415
Abstraction	1.72 (0.53)	0–2	1.70 (0.47)	1–2	1.66 (0.48)	1–2	0.8641
Delayed Recall	4.41 (0.82)	2–5	4.53 (0.63)	3–5	4.48 (0.69)	3–5	0.8142
Orientation	5.90 (0.31)	5–6	5.87 (0.35)	5–6	5.97(0.19)	5–6	0.4102
FAB total score	15.79 (1.37)	13–18	16.13 (1.55)	11–18	16.14 (1.46)	13–18	0.3696
Similarities (conceptualization)	2.21 (0.73)	0–3	2.30 (0.70)	0–3	2.31 (0.60)	1–3	0.8154
Lexical Fluency (mental flexibility)	2.76 (0.51)	1–3	2.83 (0.38)	2–3	2.69 (0.60)	1–3	0.5526
Motor Series (programming)	2.69 (0.47)	2–3	2.83 (0.53)	1–3	2.79 (0.41)	2–3	0.4916
Conflicting Instructions (sensitivity to interference)	2.72 (0.65)	1–3	2.77 (0.50)	1–3	2.79 (0.49)	1–3	0.8913
Go–No-Go (inhibitory control)	2.41 (0.78)	0–3	2.40 (0.81)	0–3	2.59 (0.82)	0–3	0.6183
Prehension Behavior (environmental autonomy)	3.00 (0.00)	3	3.00 (0.00)	3	2.97 (0.19)	2–3	0.3658
Mood State Measures							
GDS score	1.31 (1.75)	0–8	2.50 (2.00)	0–7	2.24 (2.49)	0–9	0.0976
WHO-QOL score	106.66 (10.60)	83–124	96.70 (9.75)	80–119	101.41 (11.38)	73–123	0.0756
Imitation							
Imitation gesture total score	1.66 (2.51)	0–10	1.30 (1.95)	0–6	2.03 (2.76)	0–10	0.5530
Gait Ability Measures							
Fast gait speed 10 m (s)	5.22 (1.07)	3.46–9.11	5.19 (0.72)	3.47–6.36	5.03 (0.79)	3.09–6.54	0.4188
Step (cm)	88.69 (13.83)	62.40–139.80	88.57 (13.13)	71.30–123.20	87.93 (10.06)	66.10–109.00	0.9690
RMS	1.17 (0.33)	0.37–1.78	1.27 (0.40)	0.42–1.93	1.32 (0.46)	0.49–2.21	0.3414
Walking cycles	0.05 (0.05)	0.00–0.26	0.04 (0.03)	0.00–0.14	0.05 (0.03)	0.01–0.13	0.5262
Physical Function Measures							
Hand grip dominant average (kg)	29.42 (9.08)	15.15–53.60	26.54 (9.24)	6.50–40.05	26.84 (8.26)	9.70–42.00	0.3987
Toe strength max average (kg)	15.37 (5.62)	3.10–26.80	15.92 (6.14)	4.20–28.90	17.21 (5.14)	8.60–29.10	0.2143
Feet side-by-side (s)	10.00 (0.00)	10	10.00 (0.00)	10	10.00 (0.00)	10	
Semi-Tandem stance (s)	10.00 (0.00)	10	10.00 (0.00)	10	10.00 (0.00)	10	
Tandem stance (s)	10.00 (0.00)	10	10.00 (0.00)	10	10.00 (0.00)	10	
One-leg stand max average (s)	75.23 (40.89)	10.59–120.00	84.84 (40.88)	3.50–120.00	82.75 (44.94)	7.00–120.00	0.4981
FTSST (s)	9.22 (2.93)	5.50–18.10	8.65 (2.51)	4.20–14.92	8.65 (2.57)	4.80–14.90	0.4112
Shoulder Flexion Active dominant (°)	158.10 (12.85)	115–175	163.00 (8.05)	145–180	160.00 (9.16)	145–180	0.1834
Heel up (°)	37.40 (4.77)	26.57–50.19	38.65 (6.01)	12.43–45.00	39.03 (3.58)	29.36–46.02	0.4199
Toe up (°)	13.54 (4.95)	3.81–25.64	13.88 (4.15)	1.91–20.56	14.52 (3.88)	4.16–21.16	0.6824
Body Composition Measures							
Body height (cm)	166.13 (7.35)	153.00–179.50	162.77 (9.00)	146.40–179.00	164.09 (7.93)	148.00–177.00	0.3427
Body weight (kg)	63.9 (12.90)	42.30–96.00	60.62 (12.73)	40.00–92.20	60.80 (10.07)	41.80–78.70	0.3255
Body muscle mass (kg)	45.80 (8.29)	34.10–68.90	43.25 (8.91)	28.90–59.60	44.22 (7.10)	32.20–58.20	0.4622
Fat mass (kg)	15.29 (6.36)	3.70–35.2	14.73 (6.89)	3.90–28.9	13.86 (5.07)	4.50–24.60	0.3764
Body ECW/TBW	0.38 (0.01)	0.37–0.39	0.38 (0.00)	0.38–0.39	0.38 (0.01)	0.37–0.40	0.5532
Phase Angle	5.23 (0.62)	3.74–6.46	5.12 (0.52)	4.23–6.16	5.12 (0.60)	3.70–7.05	0.7242
Protein (kg)	9.52 (1.75)	7.10–14.30	8.98 (1.87)	6.00–12.3	9.20 (1.50)	6.70–12.40	0.4733
BMI (kg/m^2^)	23.05 (3.68)	15.80–31.80	22.74 (3.48)	16.40–29.80	22.49 (3.06)	17.20–28.90	0.5277
SMI (kg/m^2^)	7.33 (1.02)	5.80–10.10	7.05 (1.10)	5.40–9.00	7.13 (0.86)	5.20–8.70	0.4733

Values are presented as means (SDs) unless stated otherwise. SD, standard deviation; ANOVA, analysis of variance; MMSE, Mini-Mental State Examination; IADL, Instrumental Activities of Daily Living; Weight Loss, weight loss of 5% or more over the previous 2 years; A Felling of Fatigue, GDS-15 sub-item “Do you feel that you are full of vitality?”; OA, osteoarthritis; ACL, anterior cruciate ligament injury; MoCA, Montreal Cognitive Assessment; FAB, Frontal Assessment Battery at Bedside; GDS, Geriatric Depression Scale; WHOQOL-26, World Health Organization Quality of Life Scale-26; RMS, root mean square; FTSST, Five Times Sit-to-Stand Test; ECW/TBW, ratio of extracellular water/total body water; BMI, body mass index; SMI, skeletal muscle mass index.

**Table 2 ijerph-19-06202-t002:** Comparison of the changes in scores between the three groups following the 4-week intervention.

Criterion	Walking Group		Dance Group		Control Group		*F* Value	ANCOVA Permutation *p*-Value	Adjusted by FDR *p*-Value	Effect Size (*η*^2^)	Post hoc Scheffé Test		
	(*n* = 27)		(*n* = 30)		(*n* = 29)						Walking vs. Control *p*-Value (95% CI)	Dance vs. Control *p*-Value (95% CI)	Walking vs. Dance *p*-Value (95% CI)
	**Mean**	**SD**	**Mean**	**SD**	**Mean**	**SD**							
Cognitive Function Measures													
MoCA total score	0.7037	1.9178	2.0667	1.8557	−0.2414	2.1983	9.681	0.0000	0.0000 *	0.19	0.1233 (2.4910–−0.2350)	0.0000 (1.2007–3.4154) *	0.0135 (−2.4910–−0.2350) *
Visuospatial/Executive	−0.0741	0.8738	0.5333	0.9371	−0.2069	1.1765	4.518	0.0074	0.0280 *	0.10	0.8413 (−0.4302–0.6958)	0.0050 (0.1920–1.2885) *	0.0296 (−1.1659–−0.0489) *
Naming	0.0370	0.1925	0.0000	0.0000	−0.0690	0.3714	3.088	0.0819	0.1455	0.07			
Attention	−0.1852	0.8338	0.3000	0.5350	−0.2069	0.6750	2.875	0.0740	0.1398	0.07			
Language	0.7037	0.9121	0.4333	1.1943	0.2069	0.7736	3.456	0.0040	0.0170 *	0.08	0.0271 (0.0458–0.9478) *	0.4411 (−0.2128–0.6656)	0.3262 (−0.1770–0.7178)
Abstraction	0.0370	0.6493	0.2000	0.4842	0.1379	0.4411	1.599	0.1965	0.2784	0.04			
Delayed Recall	0.1852	1.0014	0.3667	0.6687	0.2414	0.9124	1.903	0.0856	0.1455	0.05			
Orientation	0.0000	0.3922	0.0667	0.3651	−0.3103	0.6603	3.521	0.0138	0.0427 *	0.08	0.0256 (0.0309–0.5897) *	0.0038 (0.1049–0.6491) *	0.8356 (−0.3438–0.2105)
FAB total score	0.2963	1.0309	0.7333	1.5522	−0.5862	1.6801	7.211	0.0004	0.0034 *	0.15	0.0369 (0.0432–−1.7218) *	0.0006 (0.5022–−2.1369) *	0.4282 (−1.2600–0.3955)
Similarities (conceptualization)	0.0741	0.6156	0.0000	0.8305	0.0000	0.4629	0.001	1.0000	1.0000	0.00			
Lexical Fluency (mental flexibility)	0.0741	0.6752	0.1333	0.3457	0.0345	0.6805	1.450	0.1604	0.2371	0.03			
Motor Series (programming)	−0.2963	0.8689	0.0333	0.7184	−0.2759	0.8822	3.175	0.0414	0.1005	0.07			
Conflicting Instructions (sensitivity to interference)	0.1481	0.6624	0.2000	0.5509	0.0690	0.5299	1.353	0.4661	0.6052	0.03			
Go–No-Go (inhibitory control)	0.2963	0.9929	0.3667	0.8503	−0.4138	1.1807	4.672	0.0128	0.0427 *	0.10	0.0060 (0.1736–1.2466) *	0.0017 (0.2580–1.3029) *	0.9471 (−0.6025–0.4618)
Prehension Behavior (environmental autonomy)	0.0000	0.0000	0.0000	0.0000	0.0000	0.2673	0.996	0.8945	0.9811	0.02			
Mood State Measures													
GDS score	−0.1481	1.5116	−1.0000	1.7019	−0.5862	1.5004	0.625	0.6154	0.7215	0.02			
WHO-QOL score	0.0370	5.9387	2.4667	8.5772	−1.8966	4.9594	2.849	0.0282	0.0799 †	0.07	0.5388 (−2.3865–6.2537)	0.0402 (0.1564–8.570) *	0.3724 (−6.7149–1.8556)
Imitation													
Imitation gesture total score	1.5556	2.4859	−0.6333	2.0424	0.5517	2.8609	2.849	0.0000	0.0000 *	0.23	0.1834 (−0.3412–2.3489)	0.0847 (−2.4948–0.1247)	0.0005 (0.8547–3.5231) *
Gait													
Fast gait speed 10 m (s)	−0.0200	0.7200	−0.5450	0.5807	−0.3066	0.5805	4.760	0.0036	0.0170 *	0.11	0.1820 (−0.0958–0.6637)	0.2798 (0.6082–0.1313)	0.0038 (0.1457–0.8991) *
Step (cm)	−2.0704	12.5372	−1.9967	10.5704	−1.0483	13.2590	0.075	1.0000	1.0000	0.00			
RMS (1/m)	−0.0304	0.2789	−0.2048	0.3577	−0.2384	0.4310	0.198	0.1257	0.1943	0.04			
Walking cycles	−0.0128	0.0459	0.0125	0.0403	0.0106	0.0389	2.902	0.0322	0.0842 †	0.07	0.0132 (−0.0426–−0.0041) *	0.9658 (−0.0168–0.0207)	0.0059 (−0.0444–−0.0062) *
Motor Function Measures													
Hand grip dominant average (kg)	0.5574	4.5520	1.8067	6.7980	0.6069	3.2131	1.951	0.0511	0.1086	0.06			
Toe strength max average (kg)	2.7600	3.1700	2.1700	3.9100	1.9500	2.9300	0.446	0.4904	0.6052	0.01			
One-leg stand max average (s)	19.6600	29.2000	9.0700	22.8000	3.4400	14.0200	6.935	0.0006	0.0041 *	0.08	0.0186 (2.2494–30.1941) *	0.5892 (−7.9775–19.2341)	0.1690 (−3.2662–24.4531)
FTSST (s)	−0.0226	0.7199	−2.5300	2.1900	−1.6200	2.0700	3.100	0.0568	0.1136	0.07			
Shoulder Flexion Active dominant (°)	5.7407	12.3805	5.0000	8.9056	1.5517	7.0841	1.662	0.1214	0.1943	0.04			
Heel Lift (°)	1.6825	2.9691	2.1503	5.3196	−0.5263	2.2597	5.904	0.0000	0.0000 *	0.13	0.0083 (0.4832–3.9343) *	0.0008 (0.9963–4.3569) *	0.7930 (−2.1795–1.2438)
Toe Lift (°)	1.3790	3.9333	2.7775	3.7821	0.4742	4.0792	2.805	0.0490	0.1086	0.07			
Body Composition Measures													
Body weight (kg)	−0.0667	1.1774	0.1533	0.7592	0.1069	0.8242	0.457	0.9263	0.9842	0.01			
Body muscle mass (kg)	−0.5370	1.2013	−0.1833	0.7391	−0.2276	0.9640	1.065	0.2898	0.3941	0.03			
Fat mass (kg)	0.4630	1.2564	0.3067	0.8354	0.3414	0.8060	0.565	0.8916	0.9811	0.01			
Body ECW/TBW	−0.0005	0.0025	−0.0007	0.0028	0.0010	0.0025	4.658	0.0010	0.0057 *	0.11	0.0859 (−0.0032–0.0001)	0.0356 (−0.0034–−0.0001) *	0.9498 (−0.0015–0.0019)
Phase Angle (°)	−0.0700	0.2000	−0.0700	0.3300	−0.1000	0.2800	0.565	0.4984	0.6052	0.01			

Values are presented as means (SDs) unless stated otherwise. SD, standard deviation; ANCOVA, analysis of covariance; FDR, Storey’s false discovery rate; 95% CI, 95% confidence interval; MoCA, Montreal Cognitive Assessment; FAB, Frontal Assessment Battery at Bedside; GDS, Geriatric Depression Scale; WHOQOL-26, World Health Organization Quality of Life Scale-26; RMS, root mean square; FTSST, Five Times Sit-to-Stand Test; ECW/TBW, ratio of extracellular water/total body water. * *p* < 0.05, † 0.05 ≤ *p* > 0.10.

## Data Availability

The raw data supporting the conclusions of this article will be made available by the authors without undue reservation.

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
