# Peer review of "Effects of Two Short-Term Aerobic Exercises on Cognitive Function in Healthy Older Adults during COVID-19 Confinement in Japan: A Pilot Randomized Controlled Trial"

_ijerph, 2022, doi:10.3390/ijerph19106202_

Round 1

Reviewer 1 Report

Dear authors
In general, it is a clear and well-reasoned document.
Here I put some suggestions that will undoubtedly improve the manuscript.
The title is very long. It is unnecessary to mention everything that has been done, only the most important. Something like: Effects of two short-term aerobic exercises on cognitive function in healthy older adults during COVID-19 confinement in Japan: A pilot randomized controlled trial.
The introduction and problem are clear and well supported.
Due to the COVID-19 pandemic, many methodologically incomplete, biased, and unreliable works have been published in the last two years. In this sense, the authors must justify the validity of their work based on their methodology and, in their results, calculate the size of the treatment effect under the effects of the pandemic vs. others already published in other conditions. The justifications seem fair but scientifically unacceptable.
The methodology is extensive, and the statistics are correct and well applied.
Table 1. The root mean square (RMS) of what variable is speaking?
As mentioned above, how the authors ensured that the participants complied with their treatment sessions and that they did not make up the numbers and information submitted. Something like observers or witnesses, recording of video sessions, accelerometry and heart rate. Without any record of this type, the information presented loses validity.
Line 262 mentions that the NW program served 90 participants. The sample size is shown in Figure 1 but is not mentioned in Session 2.6. Please clarify.
Lines 247-255. How long were the NW sessions? At the beginning of the paragraph, it is mentioned 60-min sessions (line 147), and later (251-253), it says 45-min sessions. The same thing happens with dance sessions. In line 280, it is mentioned four sessions of 30 min per week; but in lines 283-297, it says three sessions of 45 min per week.
According to paragraphs 299-310, the participants who performed physical exercise consumed twice the food provided (before and after training). Instead, the control group once (10:00 or 15:00 hours). Please explain this difference between groups and how this may affect your protocol and results.
In a research study, it is incomprehensible to say that the participant was given "a protein supplement containing approximately 8.0 g of protein". Do you not know the exact quantity and composition of the food? At least of each macronutrient. In addition, for the age of the participants, it is also essential to know what their glycemic index was? Did they carry out the respective proximal analyzes of the food?
The inclusion and exclusion criteria of the participants should be explained in the first section of the methods.
If the authors had multifrequency bioelectrical impedance equipment, I don't understand why was chose the 50 kHz frequency, which measures only extracellular body water. In addition, the distribution of body water changes with position; therefore, the subject must be in the measurement position for at least 5 min before the evaluation.
The authors must mention the chair's material, where the participants were evaluated. Also, the position of the lower and upper limbs; because must be separated from the body during the measurements.

Author Response

Dear Reviewer 1:

We thank you and Reviewer 1 for your thoughtful suggestions and insights, which have enriched the manuscript and produced a better and more balanced account of the research. We hope that the revised manuscript is now suitable for publication in your journal.

Thank you for your consideration. We look forward to hearing from you.

Reviewer 2 Report

Congratulations to the authors for conducting a clinical trial in the midst of a pandemic. It is an extra effort to conduct this type of study at this time.

As general considerations:
- The manuscript should be redone throughout. It is excessively long. It is recommended that the authors summarize all the sections much more. 

I attach the document with a series of special considerations.

Author Response

Dear Reviewer 2:

The authors would like to thank reviewer 2 for your constructive critique to improve the manuscript. We have made every effort to address the issues raised and to respond to all comments. Please, find next a detailed, point-by-point response to the reviewer's comments. Also, I have shortened the manuscript on your advice. We hope that our revisions would meet the reviewer’s expectations.

Thank you for your consideration.
Please see the attachment.

Reviewer 3 Report

The authors provide new evidences that the short-term (4 weeks) aerobic exercises Nordic working and dance improve physical function in aging adults ( 60 years). They go on to show that the dance training has additional improvement in cognitive function.  They conclude that dance is effective in improving cognitive function in the older adults.

To my knowledge, this manuscript might be the first study to compare the physical and cognitive effects between Nordic working and dance in older adults. The results may particularly benefit aging adults to improve their health during the COVID-19 pandemic.

Author Response

We would like to thank the reviewer for the positive evaluation of our work.

Round 2

Reviewer 1 Report

Suggestions were properly addressed

Reviewer 2 Report

The authors have made all the necessary changes